# Emerging Role of Decoy Receptor-2 as a Cancer Risk Predictor in Oral Potentially Malignant Disorders

**DOI:** 10.3390/ijms241814382

**Published:** 2023-09-21

**Authors:** Lucas de Villalaín, Saúl Álvarez-Teijeiro, Tania Rodríguez-Santamarta, Álvaro Fernández del Valle, Eva Allonca, Juan P. Rodrigo, Juan Carlos de Vicente, Juana M. García-Pedrero

**Affiliations:** 1Department of Oral and Maxillofacial Surgery, Hospital Universitario Central de Asturias and Instituto de Investigación Sanitaria del Principado de Asturias (ISPA), 33011 Oviedo, Asturias, Spain; lvillalain@hotmail.com (L.d.V.); taniasantamarta@gmail.com (T.R.-S.); alvarofdelvalle@hotmail.com (Á.F.d.V.); 2Department of Otolaryngology, Hospital Universitario Central de Asturias and Instituto de Investigación Sanitaria del Principado de Asturias (ISPA), 33011 Oviedo, Asturias, Spain; saul.teijeiro@gmail.com (S.Á.-T.); ynkc1@hotmail.com (E.A.); jprodrigo@uniovi.es (J.P.R.); 3Instituto Universitario de Oncología del Principado de Asturias, Universidad de Oviedo, 33006 Oviedo, Asturias, Spain; 4CIBER de Cáncer (CIBERONC), Instituto de Salud Carlos III, 28029 Madrid, Spain

**Keywords:** oral potentially malignant disorders, oral leukoplakia, senescence, DcR2, DEC1, Ki67, cancer risk marker

## Abstract

The aim of this study was to evaluate the expression of the senescence markers, Decoy Receptor 2 (DcR2) and Differentiated Embryo-Chondrocyte expressed gen 1 (DEC1), in oral potentially malignant disorders (OPMDs) to ascertain their possible association with oral cancer risk. The immunohistochemical analysis of DcR2 and DEC1 expression (along with p16 and Ki67 expression) was carried out in 60 patients with clinically diagnosed oral leukoplakia. Fifteen cases (25%) subsequently developed an invasive carcinoma. Correlations between protein marker expression, histological grade and oral cancer risk were assessed. DcR2, DEC1 and Ki67 protein expressions were found to correlate significantly with increased oral cancer risk, and also with an increased grade of dysplasia. Multivariate analysis demonstrated that DcR2 and Ki67 expression are independent predictors of oral cancer development. Our results evidence for the first time the potential of DcR2 as an early biomarker to assess oral cancer risk in patients with oral leukoplakia (HR = 59.7, *p* = 0.015), showing a superior predictive value to histology (HR = 4.225, *p* = 0.08). These findings reveal that the increased expression of DcR2 and DEC1 occurred frequently in OPMDs. In addition, DcR2 expression emerges as a powerful biomarker for oral cancer risk assessment in patients with oral leukoplakia.

## 1. Introduction

Oral cancer is the most common malignancy of the head and neck region among all malignant tumors developed from the oral mucosa, with oral squamous cell carcinoma (OSCC) being the most common neoplasm [1,2]. The term “oral potentially malignant disorders” (OPMDs) includes a number of potentially cancerous mucosal lesions, oral leukoplakia being the most frequent premalignancy [3]. According to the World Health Organization (WHO), OPMDs have characteristic presentations, and epithelial dysplasia may or may not be present [1]. A binary system has been advocated for grading oral epithelial dysplasia (OED), but it requires validation before it can be routinely applied in the oral cavity [1]. Thus, according to the WHO [1], OEDs are currently divided into the three traditional grades of severity (mild, moderate and severe dysplasia). The risk of malignant transformation varies widely between 6.6% and 36.4%, with a latency period that can reach up to 30 years [2].

The concept of cellular senescence was initially applied to irreversible and permanent cell cycle arrest after prolonged in vitro replication [4]. However, this notion has recently evolved to include the ultimate and irreversible loss of cellular replicative capacity triggered by numerous causes, such as oxidative stress, telomere dysfunction and DNA damage. In this regard, tumor cells are subject to multiple stress signals that can trigger cell senescence, such as the activation of oncogenes, a loss of tumor suppressor genes and chemo or radiotherapy treatments. In cancer, cellular senescence is considered a double-edged sword with contradictory effects: either promoting cancer development or avoiding the malignant transformation of tumor cells. Nevertheless, at present, it is well known that senescence represents an important anti-proliferative mechanism which may act as a potent anti-tumorigenic barrier that needs to be overcome in the early stages of cancer development [5,6,7]. There are numerous molecular markers associated with senescence programs, although none of them are currently considered very specific. Therefore, the proper identification of senescent cells will require the use of several markers simultaneously, along with proliferation markers [7,8].

Two of the most well-known proliferation and cell cycle markers used today in immunohistochemistry are the proteins Ki67 and p16. Ki67 is a nuclear cell cycle associated protein whose expression is strictly related to cell proliferation and also to the severity of OPMDs [9,10,11,12,13]. p16, the product of the *CDKN2* gene (cyclin-dependent kinase number 2), binds to CDK4 (cyclin-dependent kinase number 4), which inhibits cell proliferation [14]. Several groups have reported that the loss of p16 expression occurs in early stages of oral cancer [12,14,15].

TRAIL (tumor necrosis factor-related apoptosis-inducing ligand) is a type 2 transmembrane protein that selectively induces apoptosis in tumor cells but not in normal cells [16]. TRAIL interacts with four known receptors: two pro-apoptotic (DR4 and DR5) and two potentially anti-apoptotic proteins lacking the death domains (DcR1 and DcR2), which work together and balance the signaling of cell apoptosis [17,18]. Consequently, the expression levels of DcR2 have been proposed as a molecular marker of cellular senescence [6,8,19]. Furthermore, a reduction in DcR2 expression compared with normal mucosa has been reported in oral cancer [20,21].

DEC1 (differentiated embryo-chondrocyte expressed gen 1) is a transcription factor that belongs to a subfamily of basic helix–loop–helix transcription factors. DEC1 is expressed in most adult and embryonic tissues, as well as in several human tumors including oral cancer [22,23,24,25,26,27], and acts as a transcription repressor that regulates cell cycle, differentiation and apoptosis in response to various stimuli [23,28]. Depending on the tissue and cellular context, DEC1 may have pro-apoptotic or pro-survival activities [29]. DEC1 has also been considered a marker of senescence and it is overexpressed in premalignant lesions [5,6,8].

Hence, the overall goal of this work was to investigate the clinical significance of the senescence markers DcR2 and DEC1 in the early stages of oral tumorigenesis. To this purpose, protein expression analysis was performed using immunohistochemistry in a selected cohort of 60 patients with histopathological diagnoses of oral leukoplakia, and subsequently their potential predictive value for the risk of progression to OSCC was assessed.

## 2. Results

### 2.1. Patient Characteristics

Sixty patients who met the above-described inclusion criteria were enrolled in this study. Thirty (50%) of the sixty patients included in this study were men, and the remaining thirty were women, with a mean age of 60.17 years (SD 15.34, range 18–87). Ten (16.7%) patients were or had been regular alcohol drinkers and thirteen (21.7%) patients were or had been regular smokers. The mean tobacco consumption was 20 cigarettes a day. Forty-seven of the sixty analyzed white mucosal lesions (78.3%) were classified as squamous hyperplasia, five (8.3%) as mild dysplasia, three (5%) as moderate dysplasia and five (8.3%) as severe dysplasia. During the follow-up period, 15 (25%) patients developed an invasive OSCC. There was a statistically significant correlation in the present cohort between the histopathological grade and the risk of progression to oral cancer (Table 1).

### 2.2. Ki67, p16, DcR2 and DEC1 Expression in Oral Tumorigenesis

Firstly, the expression levels of Ki67, p16, DcR2 and DEC1 were evaluated in normal epithelia and only Ki67 showed a weak staining restricted to the basal layer, whereas the immunoexpression of the other markers tested was negligible.

Ki67 expression in the upper two thirds of the epithelium (positive) was observed in 12 (25.5%) squamous hyperplasias without dysplasia. The proportion increased to 76.9% of dysplastic lesions (Figure 1A,B). Positive Ki67 expression correlated significantly with malignant transformation: twelve (54.5%) of the twenty-two Ki67-positive premalignant lesions evolved to carcinoma, compared with three (8%) of the thirty-eight lesions that showed negative Ki67 expression (*p* < 0.0001; Table 1). We also found that Ki67 immunoexpression increased with the grade of dysplasia (*p* = 0.001; Table 2). The degree of Ki67 expression, measured as the percentage of stained nuclei, also increased in dysplastic lesions compared with hyperplastic epithelia (*p* = 0.001; Table 2), and correlated significantly with progression to invasive carcinoma (*p* < 0.0001) (Table 1).

Only eight (13.3%) of sixty oral premalignant lesions showed positive p16 immunostaining (>10% cells). The staining was predominantly nuclear (Figure 1C,D). p16 positivity was not associated with the histopathological diagnosis, since it was observed in six/forty-seven (13%) hyperplastic lesions and in two/thirteen (15%) dysplasias (*p* = 0.28; Table 2). In addition, p16 expression was not found to correlate with malignant transformation (*p* = 0.68; Table 1).

A moderate to strong epithelial expression of DcR2 (Figure 2A) was significantly correlated with the risk of progression to oral cancer (*p* < 0.0001; Table 1). We also observed a statistically significant association between epithelial DcR2 expression and the grade of dysplasia (*p* = 0.003; Table 2). DcR2 expression was also found in the stroma surrounding the epithelial lesions, and, consequently, DcR2 expression in this compartment was independently analyzed and scored (Figure 2B,C). Interestingly, a moderate to strong stromal expression of DcR2 was significantly associated with malignant transformation (*p* = 0.039; Table 1).

Nuclear DEC1 expression in the suprabasal layers of the epithelium (score 2; Figure 2D) was found to correlate significantly with the risk of progression to invasive carcinoma (*p* = 0.001; Table 1), and with the histological grade of dysplasia (*p* = 0.045; Table 2). In addition, cytoplasmic DEC1 expression in the epithelium was also independently analyzed and scored (Figure 2D). In good agreement with our findings for nuclear expression, cytoplasmic DEC1 expression in the suprabasal layers was significantly associated with the risk of malignant transformation (*p* < 0.0001; Table 1), and with a higher grade of dysplasia (*p* = 0.005; Table 2).

Univariate Cox analysis showed that the presence of dysplasia, moderate to strong epithelial DcR2 expression, moderate to strong Ki67 expression, suprabasal Ki67 expression and cytoplasmic DEC1 expression above the basal layer were significantly associated with oral cancer risk (Table 3), thus exhibiting shorter time to progression to OSCC (Table 3). All these factors were included in a multivariate Cox regression model. In this analysis, epithelial DcR2 and Ki67 expression were significant independent predictors of oral cancer development (HR = 59.7 and HR = 4.14, respectively) (Table 4).

We also repeated the analyses using the binary grading classification (low-grade vs. high-grade dysplasia) (Appendix A). Concordantly, Ki67, DcR2 and DEC1 showed significant associations with oral cancer risk.

### 2.3. Ki67, p16, DcR2 and DEC1 Expression in Oral Cancer

We also extended the immunohistochemical analysis of these markers to the 15 OSCC developed in our series of patients with OPMDs. In general, we observed an increased expression of these markers in oral carcinomas, except for epithelial DcR2 expression. Absolute and relative frequencies of marker expression are shown in Table 5. It is noteworthy that the elevated expression of DcR2 was also observed in the peritumoral stroma (Figure 2C).

## 3. Discussion

This study investigates the role of DcR2 and DEC1, two markers of cellular senescence [5,6,8,19], in the early stages of oral tumorigenesis and malignant transformation. To accomplish this, immunohistochemical expression analysis was carried out in a large series of OPMDs with varying degrees of epithelial dysplasia to establish correlations with clinicopathological data and oral cancer risk. Protein expression was also assessed in the invasive tumors subsequently developed. Additionally, the proliferative marker Ki67 and cell cycle marker p16 were included in the analysis.

Our results demonstrate that DcR2 immunoexpression, in both the epithelium and the stroma, underneath the premalignant lesions, increases during oral carcinogenesis. More importantly, DcR2 expression correlates significantly with the risk of progression to invasive carcinoma, which shows DcR2’s potential as a predictive marker for malignant transformation. Increased DcR2 expression has also been reported in tumors from other locations [30]. Furthermore, recent articles have postulated that air pollution may play a role in the etiopathogenesis of OPMDs by increasing oxidative stress and creating an inflammatory environment [31]. Oxidative stress also plays an important role in the senescence process through several mechanisms [32]. In this scenario, DcR2 could also emerge as an important mediator of the development of OPMDs in response to air pollution.

TRAIL can induce apoptosis in tumor cells, independently of p53, without damaging normal cells [16]. However, a study from Sancilio and co-workers demonstrated that the binding of TRAIL to DcR2 could provide anti-apoptotic signals via the stimulation of the NF-kB pathway [33]. Thus, increased DcR2 expression could be considered a reflection of anti-apoptotic activity, and therefore, DcR2 may be playing a pro-oncogenic role favoring tumor growth. Additionally, although there is sufficient evidence of antitumor activity of compounds called PARAs (pro-apoptotic receptor agonists) [34,35], their clinical use has not shown the desired results [36], and one possible explanation for this lack of efficacy may be this anti-apoptotic effect exerted by DcR2 [34]. Furthermore, it has recently been described that the hypermethylation of the *DcR2* gene is associated with OSCC occurrence and development, pointing to the methylation as a new target in the therapy of OSCC [17].

Our results also revealed that DcR2 expression in the stromal compartment may play an important role in the early stages of oral carcinogenesis, and that stromal senescence may act in a tumor-promoting way. We have detected moderate DcR2 expression in six (40%) and strong expression in seven (46.7%) peritumoral stroma out of fifteen squamous carcinomas developed in our series. In this regard, some studies have also shown that senescent cells may have deleterious effects on the tissue microenvironment [37,38], such as the acquisition of a senescence-associated secretory phenotype (SASP). SASP encompasses foremost pro-inflammatory cytokines but also proteins related to extracellular matrix and cell division [39], which shows that senescent fibroblasts in the stroma possess the ability to stimulate premalignant epithelial cells to proliferate [40,41].

The knowledge of the expression profile of TRAIL receptors in a particular kind of tumor could help to predict biological behavior and response to pro-apoptotic drugs [42]. Nevertheless, our results do not point to DcR2 immunoexpression as a marker of effective senescence. However, our findings do support the role of DcR2 expression as a marker of oral cancer risk.

Immunohistochemical DEC1 expression has been previously studied in tumors from different locations and histology [22,23,24,28], including some reports in OSCC that describe its role and assess the clinical value of DEC1 expression [24,26,27,43]. Liudi et al. [44] reported a positive correlation between the expression of DEC1 and HIF-1α, and showed that DEC1 expression was upregulated by hypoxia, thereby leading to the increased motility of OSCC cells. You et al. [25] found significantly higher DEC1 expression in OSCC samples than in the normal group, which was associated with early recurrence (first year), so DEC1 was considered a promoter of tumor invasion and metastasis. Meanwhile, Bhawal et al. [22] detected nuclear DEC1 expression in OSCC, carcinoma in situ and dysplasia. DEC1 expression in epithelial dysplasia was predominantly parabasal, while in oral carcinomas, it showed a homogeneous pattern throughout the tumor areas. A significant association was found in DEC1 expression with low T and clinical stage, with well-differentiated tumors and an inverse association with cyclin D1. Furthermore, a recent study by Ting et al. [27] evaluated the expression of DEC1 in oral leukoplakias and OSCC and identified it as a potential biomarker of malignant transformation in the carcinogenesis of OSCC. Notably, our study is a longitudinal study, where follow up was performed over time, unlike previously published cross-sectional studies.

In short, similar to our findings for DcR2, we found that DEC1 expression was significantly associated with the histological grade of dysplasia and the risk of malignant transformation. DEC1 can be considered a cancer risk marker in patients with oral leukoplakia, probably reflecting the DNA damage that takes place in the process of oral carcinogenesis.

In addition, we have also analyzed the expression of Ki67 and p16 in the same series of patients. We observed an increased Ki67 immunoexpression in dysplastic lesions and patient-matched invasive carcinomas compared with hyperplasias without dysplasia, according to previous reports [10,11]. Our results also showed a statistically significant association between Ki67 expression and the risk of progression to oral carcinoma. Furthermore, in a multivariate analysis, Ki67 expression above the basal third of the epithelium was a significant independent predictor of oral cancer development (HR = 4.14). This suggests that Ki67 could be a good marker to predict the risk of malignant transformation in patients with OPMDs. Nonetheless, there was no statistically significant association between p16 expression and the clinical outcome.

Epithelial dysplasia has been considered an important factor in assessing the risk of malignant transformation. In this work, we found a statistically significant association between the histological grade of dysplasia and the risk of malignant transformation and a statistical relationship between the grade of dysplastic mucosal lesions and three of the four markers tested (Ki67, DcR2 and DEC1). These markers, as well as dysplasia, were significant predictors of oral cancer using univariate analysis. Furthermore, the epithelial expressions of DcR2 and Ki67 were independent predictors of oral cancer development in multivariate analysis (HR = 59.7, *p* = 0.015 and HR = 4.14, *p* = 0.02, respectively), showing superior predictive value to dysplasia (HR = 4.225, *p* = 0.08), which highlights the limited predictability of histology in patients with OPMDs.

Among the limitations of this study are those related to the sample size and retrospective design. To minimize sample and selection biases, a series of precise inclusion criteria were established and all the patients who met them were systematically enrolled. Even though patient selection was retrospective, subsequent analyses were prospectively performed. Nevertheless, further validation in future prospective, large sample-size studies are needed to confirm these findings.

## 4. Materials and Methods

### 4.1. Patients and Tissue Specimens

Surgical tissue specimens from patients who were diagnosed with oral mucosa leukoplakia at the Hospital Universitario Central de Asturias between 2000 and 2005 were retrospectively collected and prospectively analyzed, in accordance with approved institutional review board guidelines. Informed consent was obtained from all patients. All experimental procedures were conducted in accordance with the Declaration of Helsinki, and approved by Institutional Ethics Committee of the HUCA and by the Regional Ethics Committee from Principado de Asturias (date of approval 14 May 2019; approval number 136/19, for the project PI19/01255). Patients with oral leukoplakia included in this study had to meet the following criteria: (i) pathological diagnosis of oral epithelial hyperplasia or dysplasia; (ii) no previous history of head and neck cancer; (iii) complete excisional biopsy of the lesion; (iv) a minimum follow-up of 5 years (or until progression to malignancy occurred); and (v) signed informed consent to use their tissues for investigation. The sections selected for study also contained normal epithelia as internal controls. Patients were followed up every 2 months for the first 6 months after completing the treatment, every 3 months until the second year, and every 6 months thereafter. Apart from excisional biopsies performed during the follow-up period, these patients were not subjected to any other type of specific treatment.

Representative tissue sections were obtained from archival, paraffin-embedded blocks. OPMDs were classified into the categories of squamous cell hyperplasia without dysplasia, mild, moderate or severe dysplasia, following the World Health Organization classification [1]. Tumor blocks were also obtained from those patients who developed an invasive OSCC. Alveolar mucosa obtained from unerupted third molar surgery was used as control. All patients gave their consent to excise this normal tissue. Tissue specimens were provided by the Principado de Asturias BioBank (PT17/0015/0023), which is part of the Spanish National Biobanks Network.

### 4.2. Immunohistochemistry

Then, 3 µm-thick tissue sections were cut and dried on Flex IHC microscope slides (Dako, Glostrup, Denmark). The sections were deparaffinized and antigen retrieval was performed using Envision Flex Target Retrieval solution (Dako, Glostrup, Denmark), either high pH (for p16, DcR2 and DEC1) or low pH (for Ki67). Staining was performed at room temperature on an automatic staining workstation (Dako Autostainer Plus, Dako) using the Dako EnVision Flex + Visualization System (Dako Autostainer). The antibodies and dilutions used are shown in Appendix A. Negative controls, consisting of slides stained with omission of the primary antibody, were also included. Tonsil, cervix carcinoma, kidney and breast carcinoma samples were used as positive controls for Ki67, p16, DcR2 and DEC1, respectively.

The slides were viewed randomly, without clinical data, by three of the authors, with a high level of inter-observer concordance (>95%). We undertook two separate evaluations for Ki67, which is previously described in the literature [9,11]. Positive value was assigned (1) if there was nuclear staining in the upper two thirds of the epithelium and negative (0) when the nuclear staining only was noticed in the epithelial basal third [9]. Moreover, a scoring system based on the percentage of stained cells was also applied, assigning mild expression (0) when the percentage of positive epithelial cells was less than 10%, moderate (1+) when it was between 10 and 50% and strong (2+) when it exceeded 50% [11]. p16 expression was scored by assigning negative value (0) when the percentage of epithelial cells with nuclear and/or cytoplasmic immunoexpression was less than 10% of the total and positive (1+) when it was equal to or greater than 10% [16]. DcR2 expression was analyzed separately in the epithelium and the stroma. In both cases, the scoring system was based on the percentage of cells with nuclear and/or cytoplasmic staining [20,21], assigning mild expression (0) when the percentage of positive epithelial or stromal cells was less than 5%, moderate (1+) when it was between 5 and 50% and strong (2+) when it exceeded 50%. DEC1 immunoexpression was evaluated independently in the nucleus and in the cytoplasm. In both cases, the scoring system applied was similar to that previously described for podoplanin [2,45]: (0) if no expression was observed in any part of the epithelium, (1+) expression restricted to the basal layer of the epithelium, (2+) expression in the basal and suprabasal layers of the epithelium.

### 4.3. Statistical Analysis

Statistical analysis was performed using IBM SPSS for Windows (version 27.0.1, IBM-SPSS Inc., Armonk, NY, USA). Chi-square and Fisher’s exact test were used for comparison between categorical variables. Analysis of survival was performed using the Kaplan–Meier method, and comparison of survival rates was performed by using the log-rank test. Cox proportional hazard models were used for univariate and multivariate analyses. The hazard ratios (HR) with their 95% confidence intervals (CI) and *p* values were reported. All tests were two-sided. *p* values ≤ 0.05 were considered statistically significant.

## 5. Conclusions

This study demonstrates that the increased expression of DcR2, DEC1 and Ki67 occurred frequently in OPMDs, and more importantly, were significantly correlated with the risk of progression to invasive carcinoma. Multivariate analysis revealed the potential of DcR2 and Ki67 as independent predictors of oral cancer development. Furthermore, DcR2 emerges as a powerful biomarker for oral cancer risk assessment in patients with oral leukoplakia, showing superior predictive value to the histological grade, which is the gold standard in clinical practice.

## Figures and Tables

**Figure 1 ijms-24-14382-f001:**
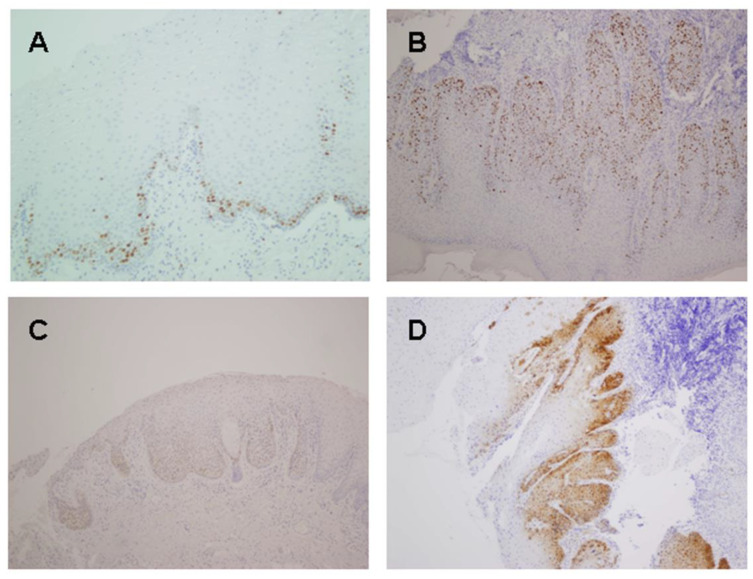
Representative examples of Ki67 and p16 immunoexpression in oral leukoplakia. Ki67 expression restricted to the basal layer in an oral leukoplakia that did not undergo malignant transformation (**A**), whereas Ki67 expression extended above the basal third of the epithelium and affected a high percentage of cells in a case of oral dysplastic leukoplakia that finally evolved to invasive carcinoma (**B**). Two cases of dysplastic oral epithelia showing p16 staining in basal and suprabasal layers of epithelium; however, subsequent evolution was different. While the case shown in (**C**) underwent malignant transformation, the case in (**D**) did not progress to oral cancer. Original magnification 200× (**A**); 100× (**B**–**D**).

**Figure 2 ijms-24-14382-f002:**
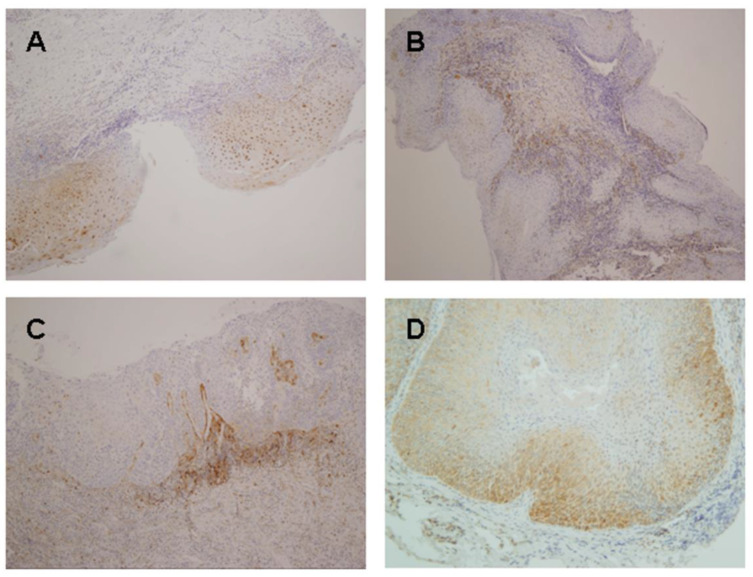
Representative examples of immunohistochemical expression patterns of DcR2 and DEC1 in oral leukoplakia and patient-matched OSCC. Examples of dysplastic lesions that evolved to invasive carcinoma showing either a distribution of DcR2 expression in the epithelium (**A**), or in the corium (**B**). Patient-matched OSCC also showed DcR2 expression in the stroma (**C**). Original magnification 100×. A patient with oral leukoplakia that subsequently developed oral carcinoma, showing both patterns of nuclear and cytoplasmic DEC1 expression in epithelial cells (**D**). Original magnification 200×.

**Table 1 ijms-24-14382-t001:** Evolution of the premalignant lesions in relation to histopathological diagnosis and Ki67, p16, DcR2 and DEC1 expression.

Characteristics	No. Cases (%)	Progression to OSCC [No. Cases (%)]	*p*
Histopathological diagnosis			
Without dysplasia	47 (78)	5 (11)	<0.0001
Mild–moderate dysplasia	7 (12)	5 (71)	
Severe dysplasia	6 (10)	5 (83)	
Ki67% expression (% of positive epithelial cells)			
Mild (score 0)	32 (53)	2 (6)	<0.0001
Moderate (score 1)	27 (45)	12 (44)	
Strong (score 2)	1 (2)	1 (100)	
Ki67 expression (epithelial distribution)			
Restricted to basal third (score 0)	38 (63)	3 (8)	<0.0001 *
Above basal third (score 1)	22 (37)	12 (55)	
p16 expression (% of positive epithelial cells)			
Negative (score 0)	52 (87)	13 (25)	0.68 *
Positive (score 1)	8 (13)	2 (25)	
Epithelial DcR2 expression (% of positive cells)			
Mild (score 0)	51 (85)	8 (16)	<0.0001
Moderate (score 1)	8 (13)	6 (75)	
Strong (score 2)	1 (2)	1 (100)	
Stromal DcR2 expression (% of positive cells)			
Mild (score 0)	35 (58)	5 (14)	0.039
Moderate (score 1)	24 (40)	10 (42)	
Strong (score 2)	1 (2)	0 (0)	
Nuclear DEC1 expression (epithelial distribution)			
No expression (score 0)	7 (12)	1 (14)	0.001
Restricted to basal layer (score 1)	31 (55)	3 (10)	
Suprabasal layer (score 2)	19 (33)	11 (58)	
Cytoplasmic DEC1 expression (epithelial distribution)			
No expression (score 0)	30 (53)	4 (13)	<0.0001
Restricted to basal layer (score 1)	19 (33)	3 (16)	
Suprabasal layer (score 2)	8 (14)	8 (100)	

Chi-square and * Fisher’s exact tests.

**Table 2 ijms-24-14382-t002:** Relationship between histological grade of epithelial dysplasia and expression of Ki67, p16, DcR2 and DEC1.

Protein Expression	Histological Grade of Epithelial Dysplasia				*p*
	Absent	Mild	Moderate	Severe	
Ki67 expression (% of positive epithelial cells)					
Mild (score 0)	30 (63.8)	2 (40.0)	0 (0.0)	0 (0.0)	0.001
Moderate (score 1)	17 (36.2)	3 (60.0)	3 (100.0)	4 (80.0)	
Strong (score 2)	0 (0.0)	0 (0.0)	0 (0.0)	1 (20.0)	
Ki67 expression (epithelial distribution)					
Restricted to basal third (score 0)	35 (74.5)	2 (40.0)	1 (33.3)	0 (0.0)	0.001
Above basal third (score 1)	12 (25.5)	3 (60.0)	2 (66.7)	5 (100.0)	
p16 epithelial expression					
Negative (score 0)	41 (87.2)	5 (100)	3 (100)	3 (60)	0.28
Positive (score 1)	6 (12.8)	0 (0)	0 (0)	2 (40)	
DcR2 expression (% of positive epithelial cells)					
Mild (score 0)	44 (93.6)	3 (60.0)	2 (66.7)	2 (40.0)	0.003
Moderate (score 1)	2 (4.3)	2 (40.0)	1 (33.3)	3 (60.0)	
Strong (score 2)	1 (2.1)	0 (0.0)	0 (0.0)	0 (0.0)	
Nuclear DEC1 expression (epithelial distribution)					
No expression (score 0)	6 (13.6)	1 (20.0)	0 (0.0)	0 (0.0)	0.045
Restricted to basal layer (score 1)	27 (61.4)	3 (60.0)	0 (0.0)	1 (20.0)	
Suprabasal layer (score 2)	11 (25.0)	1 (20.0)	3 (100.0)	4 (80.0)	
Cytoplasmic DEC1 expression (epithelial distribution)					
No expression (score 0)	26 (59.1)	1 (20.0)	1 (33.3)	2 (40.0)	0.005
Restricted to basal layer (score 1)	16 (36.4)	2 (40.0)	0 (0.0)	1 (20.0)	
Suprabasal layer (score 2)	2 (4.5)	2 (40.0)	2 (66.7)	2 (40.0)	

**Table 3 ijms-24-14382-t003:** Univariate Cox proportional hazards model to estimate oral cancer risk.

Variables	Mean Time [Months] to Progression (95% CI)	HR (95% CI)	*p*
Dysplasia No dysplasia Mild–moderate Severe	189.26 (164.21–214.31) 92.60 (21.71–163.49) 89.33 (64.04–114.62)	Reference 4.69 (1.33–16.56) 4.76 (1.34–16.82)	0.008
Ki67 expression Mild Moderate Strong	244.57 (204.32–284.82) 128.35 (92.63–164.07) 52.00 (52.00–52.00)	Reference 4.91 (1.09–22.14) 28.11 (2.22–355.10)	0.005
Ki67 epithelial distribution Basal Suprabasal	235.01 (193.26–276.77) 123.18 (86.48–159.88)	4.02 (1.12–14.41)	0.021
p16 expression Negative Positive	171.56 (131.52–211.60) 106.00 (73.66–138.33)	1.37 (0.30–6.22)	0.68
Epithelial DcR2 expression Mild Moderate Strong	197.29 (155.74–238.85) 70.88 (48.87–92.89) 57.00 (57.00–57.00)	Reference 6.47 (2.05–20.39) 13.32 (1.40–126.10)	<0.0001
Stromal DcR2 expression Mild Moderate Strong	200.90 (143.97–257.82) 130.48 (91.06–169.90) 57.00 (57.00–57.00)	Reference 2.46 (0.83–7.25) 0.00 (0.00–0.00)	0.12
Nuclear DEC1 expression Negative Basal Suprabasal	64.50 (51.76–77.23) 234.48 (191.24–277.73) 120.04 (83.46–156.63)	Reference 0.44 (0.04–4.40) 1.68 (0.21–13.44)	0.08
Cytoplasmic DEC1 expression Negative Basal Suprabasal	119.62 (93.79–145.46) 218.10 (162.93–273.26) 84.12 (49.62–118.62)	Reference 0.52 (0.09–2.93) 4.08 (1.20–13.84)	0.002

HR: Hazard Ratio; 95% CI: 95% Confidence Interval.

**Table 4 ijms-24-14382-t004:** Multivariate Cox regression for analyzed variables.

Variables	*P*	Hazard Ratio (HR)	95% Confidence Interval
Dysplasia (no vs. yes)	0.08	4.225	0.826–21.611
Ki67 expression (Upper two thirds vs. basal third)	0.02	4.14	1.19–14.39
p16 expression	0.425	0.47	0.07–2.98
Epithelial DcR2 expression	0.05		
5–50%	0.499	2.19	0.22–21.50
>50%	0.015	59.7	2.23–1595.1
Stromal DcR2 expression	0.62	1.35	0.39–4.60
Nuclear DEC1 expression	0.84	1.12	0.35–3.58
Cytoplasmic DEC1 expression	0.91	1.03	0.53–2.0

**Table 5 ijms-24-14382-t005:** Expression of Ki67, p16, DcR2 and DEC1 in patient-matched oral squamous cell carcinomas.

Variables	No. Cases (%)
Ki67 expression (% of tumor-stained cells)	
10%	4 (26.7)
10–50%	2 (13.3)
>50%	9 (60.0)
p16 expression (% of tumor-stained cells)	
<10%	10 (66.7)
≥10%	5 (33.3)
Epithelial DcR2 (% of stained cells)	
<5%	11 (73.3)
5–50%	2 (13.3)
>50%	2 (13.3)
Stromal DcR2 (% of stained cells)	
<5%	2 (13.3)
5–50%	6 (40.0)
>50%	7 (46.7)
Nuclear DEC1 expression	
Negative immunostaining	4 (26.7)
Positive immunostaining	11 (73.3)
Cytoplasmic DEC1 expression	
Negative immunostaining	3 (20)
Positive immunostaining	12 (80)

## Data Availability

Data available upon request to the corresponding author (JCV) due to privacy/ethical restrictions.

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
