# Peer review of "Emerging Role of Decoy Receptor-2 as a Cancer Risk Predictor in Oral Potentially Malignant Disorders"

_ijms, 2023, doi:10.3390/ijms241814382_

Round 1
Reviewer 1 Report
Thank you for giving me the opportunity to review this paper which aims to utilise the expression of two proteins (DcR2 and DEC1) in OPMD tissue in the prediction of progression to oral cancer.
The expression of DcR2 was found to be predictive of oral cancer development.
Major concern:
The whole study hinges on the analysis of protein expression on leukoplakia tissue sections, yet there is no objective method to assess expression.
The assessment of protein expression was assessed independently by three of the authors and inter-observer concordance was high.
Percentage of stained cells was assessed by the authors semi-quantitively - it is suggested that this should be conducted objectively using scanning software, and all the results analysed again with respect to oral cancer prediction.
In addition to presentation of the results in the tables, which is difficult for other clinicians to apply to their populations, it would be better to establish a risk calculator or scoring method should be established so that readers can apply the authors’ findings to their patients and understand the risk in their own patient populations.
Extensive English editing is required
Author Response
REVIEWER # 1
Comments and Suggestions for Authors
Thank you for giving me the opportunity to review this paper which aims to utilise the expression of two proteins (DcR2 and DEC1) in OPMD tissue in the prediction of progression to oral cancer.
The expression of DcR2 was found to be predictive of oral cancer development.
Response: We thank the reviewer for highlighting the major aim and utility of our findings.
Major concern: The whole study hinges on the analysis of protein expression on leukoplakia tissue sections, yet there is no objective method to assess expression.
Response: We fully agree that objective and reliable methods for oral cancer risk assessment are still an unmet need and of major importance for patients with OPMD.
The assessment of protein expression was assessed independently by three of the authors and inter-observer concordance was high.
Percentage of stained cells was assessed by the authors semi-quantitively - it is suggested that this should be conducted objectively using scanning software, and all the results analysed again with respect to oral cancer prediction.
Response: Protein staining was indeed quantified using scoring systems based on the percentage of positive cells, according to previous reports (references 9, 11, 16, 20, 21 for p16, Ki67 and DcR2). Probably the term “semi-quantitative” caused misunderstanding. Since DEC1 expression pattern was similar to that previously reported for podoplanin in OLP, the same scoring system was applied (references 2 and new ref. 30 by Kawaguchi et al.). Scoring was done by microscope examination of the whole tissue biopsy section by three independent observers, as described by Kawaguchi et al J Clin Oncol 2008. This warrants that protein expression assessment is performed specifically in the OPL lesion (i.e. hyperplasia and dysplastic areas excluding surrounding normal adjacent tissue). In fact, this is a method widely accepted and extensively used in IHC-based biomarker studies and also routinely by pathologists in daily clinical practice. As we stated, slides were rigorously reviewed blinded to clinical data by three of the authors. A high level of inter-observer concordance was reached (>95%), which further supports the consistency and reliability of our data. To avoid confusion, the term “semi-quantitative” has been removed in our revised version of the manuscript.
NEW REFERENCE: Kawaguchi H, El-Naggar AK, Papadimitrakopoulou V, Ren H, Fan YH, Feng L, Lee JJ, Kim E, Hong WK, Lippman SM, Mao L. Podoplanin: a novel marker for oral cancer risk in patients with oral premalignancy. J Clin Oncol. 2008 Jan 20;26(3):354-60. doi: 10.1200/JCO.2007.13.4072. PMID: 18202409
In addition to presentation of the results in the tables, which is difficult for other clinicians to apply to their populations, it would be better to establish a risk calculator or scoring method should be established so that readers can apply the authors’ findings to their patients and understand the risk in their own patient populations.
Response: Following the reviewer’s comment, data presentation has now been improved for Tables 1 and 3 to enhance the clarity of results obtained. Thus, the percentages in Table 1 have been made more intuitive to the findings, and data in Table 3 have been extended to include “mean time to progression” to OSCC. In addition, please note that results from Univariate and Multivariate Cox analyses, shown in Tables 3 and 4 include hazard ratios (HR) and 95% confidence intervals (CI). HR can actually be considered as an estimate of relative risk (i.e. risk of developing oral cancer).
Comments on the Quality of English Language
Extensive English editing is required
Response: Language revision/editing of this new version of our manuscript has been done by a native English speaker.
Reviewer 2 Report
Dear Authors,
You raised the important subject of search for novel biomarkers of cancer progression in OPMD. The manuscript is well written, however, there are two major concerns.
1) The study limitation section in the discussion is missing.
2) In the discussion, you omitted the impact of air pollution on OPMD and its biomarker concentrations. (i.e.: 10.26402/jpp.2019.1.09). Please expand on this issue.
Minor corrections:
1) Some references seem to be a little bit outdated. Please check if you use the newest study results.
2) Line 99-100: This is the result, not the method description, and should be stated as the first sentence of the Results section rather than in Materials and Methods.
3) Please remove the "Figure 1/2" signs on the graphic files.
4) Table 1, first 5 lines. What does the "p" value refer to? As we can see, the highest rate of progression to squamous cell carcinoma was present in polarly different stages (without and severe dysplasia).
Kind regards
Author Response
REVIEWER # 2
Comments and Suggestions for Authors
Dear Authors, You raised the important subject of search for novel biomarkers of cancer progression in OPMD. The manuscript is well written, however, there are two major concerns.
Response: We thank the reviewer for highlighting the quality of our study and the interest for the field.
1) The study limitation section in the discussion is missing.
Response: Study limitations have now been included at the end of the Discussion section.
2) In the discussion, you omitted the impact of air pollution on OPMD and its biomarker concentrations. (i.e.: 10.26402/jpp.2019.1.09). Please expand on this issue.
Response: We thank the reviewer for this insightful suggestion. The impact of air pollution on OPMD and stress oxidative markers are now discussed, and the suggested new reference cited in the revised manuscript.
NEW REFERENCE: Gregorczyk-Maga I, Celejewska-Wojcik N, Gosiewska-Pawlica D, Darczuk D, Kesek B, Maga M, Wojcik K. Exposure to air pollution and oxidative stress markers in patients with potentially malignant oral disorders. J Physiol Pharmacol. 2019 Feb;70(1). doi: 10.26402/jpp.2019.1.09. Epub 2019 Jun 3. PMID: 31172968.
Minor corrections:
1) Some references seem to be a little bit outdated. Please check if you use the newest study results.
Response: Some references have been updated (Refs. 26, 27 and 45), while maintaining those highly relevant original/seminal articles.
NEW REFERENCES:
[26] Mao T, Chen W, Xiong H, Wang C, Yang L, Hu X, et al. DEC1 is a potential marker of early metastasis in Oral squamous cell carcinoma. Tissue Cell. 2023;82:102094, doi:10.1016/j.tice.2023.102094.
[27] Mao T, Xiong H, Hu X, Hu Y, Wang C, Yang L, et al. DEC1: a potential biomarker of malignant transformation in oral leukoplakia. Braz Oral Res. 2020;34:e052, doi:10.1590/1807-3107bor-2020.vol34.0052.
[45] Yang L, Zeng L, Wang Z, Hu X, Xiong H, Zhang T, et al. Differentiated embryo chondrocyte 1, induced by hypoxia-inducible factor 1alpha, promotes cell migration in oral squamous cell carcinoma cell lines. Oral Surg Oral Med Oral Pathol Oral Radiol. 2022;133:199-206, doi:10.1016/j.oooo.2021.08.022
2) Line 99-100: This is the result, not the method description, and should be stated as the first sentence of the Results section rather than in Materials and Methods.
Response: We fully agree. The referred sentence has been moved to Results section, as suggested.
3) Please remove the "Figure 1/2" signs on the graphic files.
Response: We apologize. Figures 1 and 2 have been modified now removing “Figure 1&2” signs.
4) Table 1, first 5 lines. What does the "p" value refer to? As we can see, the highest rate of progression to squamous cell carcinoma was present in polarly different stages (without and severe dysplasia).
Response: Please note that Table 1 has now been modified to enhance the clarity of results achieved. Chi-square and Fisher’s exact tests were used for comparisons between categorical variables, as mentioned in Materials and Methods section. The associated p value for each comparison is shown. The number and percentage of cases that progressed to OSCC for each subgroup/stage is now indicated. Hopefully, these changes will help to visualize more easily the differences among the categories. It is now shown that while only 11% of OLP without dysplasia progressed to OSCC, the progression rates were increased to 70 and 80% for OLPs with mild/moderate and severe dysplasia, respectively.
Also, for your information, Supplementary Table S2 has been modified because to remove overlapping data already included in Table 2. Thus, Supplementary Table S2 is now precisely restricted to the relationship between the evolution and the histopathological diagnosis using WHO binary dysplasia grading.
Round 2
Reviewer 1 Report
Thank you for making changes to this manuscript which has improved readability and I am happy to recommend the paper for acceptance
Thank you for proofreading
Reviewer 2 Report
Dear Authors,
Thank you for the responses and improvements introduced into the manuscript. At this point, I do not have any further questions or requests.
Kind regards